# Cognitive and Emotional Resilience in Parents with Children with Autism Spectrum Disorder During COVID-19: The Role of Promoting Variables

**DOI:** 10.3390/jintelligence13010006

**Published:** 2025-01-06

**Authors:** Aziz Sarhani-Robles, María Guillot-Valdés, María Auxiliadora Robles-Bello, David Sánchez-Teruel

**Affiliations:** 1Medicine Faculty, University of Granada, 18016 Granada, Spain; asarhanir@correo.ugr.es; 2Department of Psychology, University of Jaen, 23071 Jaen, Spain; 3Faculty of Psychology, University of Granada, 18012 Granada, Spain; dsteruel@ugr.es

**Keywords:** resilience, positive mental health, emotion regulation, coping humour, reappraisal, social support, autism spectrum disorder, emotional intelligence

## Abstract

The pandemic resulting from the coronavirus disease (COVID-19) has entailed social and psychological consequences for the Spanish population, with children with autism spectrum disorder (ASD) being particularly vulnerable due to their genetic characteristics. The present study focuses on the efforts of parents of children with autism spectrum disorder to improve their situation during the pandemic. In particular, the objective is to identify promoting variables (subjective well-being, positive mental health, social support, humour, cognitive reappraisal, and self-esteem) and sociodemographic variables that predict resilience, marking positive coping with this adverse situation. Furthermore, the study conceptually explores the potential role of emotional intelligence in resilience-building processes. We hypothesised that higher scores in these promoting variables would predict greater resilience, with emotional intelligence potentially serving as an underlying framework. The methodology employed in this study is as follows: A cross-sectional predictive study was conducted on a sample of 799 parents using an online questionnaire administered during the social confinement resulting from the SARS-CoV-2 pandemic. The information analysed is based on data provided by the parents. Statistical methods included Student’s *t*-tests, Pearson’s correlations, and stepwise multivariate regression analysis to identify predictors of resilience. The results of the study are presented below. Significant resilience scores and resilience prediction were identified in participants based on positive mental health, emotion regulation, sense of humour, social support, age, and employment status (self-employed). Relations with emotional intelligence were identified, particularly in the domains of emotion regulation, cognitive reappraisal, and positive mental health. Discussion: The necessity of an intervention that prioritises the empowerment of resilience in the target population is substantiated. Practical implications suggest leveraging emotional intelligence strategies to enhance resilience in this population. This study highlights the importance the aforementioned variables, in addition to potential strategies for enhancing the sociodemographic circumstances of the families.

## 1. Introduction

Autism Spectrum Disorder (ASD) represents a heterogeneous set of neurodevelopmental disorders characterised by deficits in reciprocal social communication and social interactions in different contexts, along with restrictive and repetitive patterns of behaviour ([4]). The World Health Organization (WHO) estimates that the prevalence of autism spectrum disorder (ASD) is approximately 1 in 160 children ([62]).

ASD is a condition that encompasses a broad spectrum of neurodevelopmental disorders, ranging from mild to severely disabling. Its prevalence has significantly increased in recent years due to various factors, including changes in diagnostic criteria, public policies, healthcare practises, and increased awareness of its symptoms ([48]). The increased prevalence of individuals with ASD has led to a focus on a segment of the population that is directly influenced by the disorder, despite not having it themselves: their parents. A number of studies have indicated that parents of children with ASD are at an elevated risk of experiencing chronic stress, anxiety, depression, and overall mental health issues in comparison to parents of children with other disabilities ([30]). These parents may also be exposed to numerous factors that can lead to distress and anxiety. These include a perceived lack of control over the child’s behaviour, concerns about their intellectual functioning and adaptive behaviours, worries about parent–child relationships, high rates of exposure to adverse experiences, uncertainty about the future of the children, and disruptive behaviours. These are experienced by 50% of children with ASD and include disobedience, tantrums, aggressions, and self-injury ([32]; [38]). Consequently, families may experience elevated levels of stress, which can give rise to marital discord and, in some cases, divorce. Additionally, they may become socially isolated, spend more time providing care, perceive their ability to care for their children as diminished, lack adequate support services, face excessive and continuous child protection measures, and experience financial strain ([52]). This establishes a bidirectional relationship between family well-being and the child’s development. Impairment of family functioning leads to adverse outcomes, and vice versa ([38]). Consequently, the adverse effects experienced by parents of children with ASD as a result of elevated stress and anxiety levels during parenting can have a detrimental impact on the efficacy of the intervention.

In order to address the challenges faced by these families, parents develop coping strategies that are critical in navigating the stresses associated with raising children with ASD. These coping strategies can influence their resilience, defined as the capacity to maintain adaptive functioning and achieve positive psychological outcomes in the face of adversity ([47]). Four categories of coping strategies are commonly employed by parents of children with ASD. These include active avoidance, characterised by the use of substances, self-indulgence and emotional detachment; specific strategies, such as planning, problem-solving and seeking appropriate social support; positive coping mechanisms, including the use of humour, positive reframing and acceptance; and finally, religion/denial, which involves the use of religion or spirituality or the denial of the problem’s existence ([82]). Among these strategies, within positive coping, resilience is a notable factor. This involves the capacity to maintain adaptive functioning and achieve positive psychological outcomes in the face of adversity and can be enhanced through training. Resilience can be bolstered through some factors, which include self-esteem, cognitive flexibility, and a sense of humour ([17]). These factors are closely linked to emotional intelligence, particularly through skills such as emotion regulation and cognitive reappraisal, which enable parents to manage stress and reinterpret challenging situations more effectively ([15]; [53]).

In recent years, a contextual variable has been introduced into the family dynamics of this population as a result of the global pandemic caused by the SARS-CoV-2 virus. The disruption to school attendance had a direct impact on children with ASD, with the potential for long-term developmental regression, the loss of skills acquired during the school programme, and an increase in abnormal behaviours and emotional problems. Furthermore, the lockdown resulted in families being compelled to maintain a state of isolation within their domestic environments, while simultaneously attempting to reconcile the demands of work and family life. Furthermore, the accumulation of stress within families was exacerbated by a number of additional factors, including the impact of the pandemic on family finances, social isolation, increased childcare time, a lack of support services, uncertainty about the disease’s status, and the future of their children. As previously described, this results in a vicious circle in which increased parental stress can have a negative impact on the development of their children, and vice versa ([1]; [51]; [81]).

As evidenced by the findings of [11] ([11]), the concept of resilience has assumed greater significance in the wake of the prolonged exposure to adversity in the global pandemic. Resilience and its associated processes can be conceptualised through broader frameworks such as emotional intelligence ([40]). By integrating emotional intelligence into resilience models, we can better understand how parents adapt to and overcome the challenges associated with raising children with ASD, particularly during crises such as the COVID-19 pandemic ([70]). In light of the contextual stressors encountered during the pandemic, the relationship between coping strategies and resilience is increasingly critical. They observed discrepancies in the efficacy of coping strategies. Accordingly, in their model, they emphasise the necessity for flexibility, which entails attending to evolving situational demands, selecting strategies that align with these demands, monitoring the efficacy of the chosen strategies, re-evaluating changing situations, and modifying strategies as required.

A number factors corresponding to individual characteristics have been identified within the context of family resilience. These include self-esteem, cognitive flexibility, and a sense of humour ([3]). With regard to the role of humour, it has been the subject of extensive investigation and is regarded as a beneficial factor, associated with reduced stress levels, enhanced resilience, and diminished depression in caregivers ([55]). Similarly, [2] ([2]) delineated the existence of diverse resilience dimensions, which include a sense of humour and positive self-esteem.

Furthermore, [60] ([60]) delineated the attributes of resilient individuals. One such characteristic was the presence of positive emotions, particularly a sense of humour, which was neurobiologically linked to a reduction in autonomic activity and the strengthening of reward circuits. Another important factor they identified was cognitive flexibility/cognitive reappraisal. This was described as the capacity to interpret adverse events in a positive manner, ascribing to them a meaningful quality and discerning potential benefits from them. Cognitive reappraisal, a core strategy within emotional intelligence, enables parents to reframe stressful situations constructively, contributing to enhanced resilience ([26]). Furthermore, they proposed that self-esteem could be regarded as a resilience characteristic, as it facilitates the avoidance of cyclical problems over time.

Other factors that have been identified as relevant in resilience include positive mental health and subjective well-being. Various studies have demonstrated that parents’ well-being can have long-lasting implications for both them and their children ([20]; [59]; [66]). In particular, [31] ([31]) highlighted a negative correlation between resilience and general psychological distress, as well as other negative mental health variables, including depression, anxiety, and stress. Similarly, [81] ([81]) observed that lower resilience, the utilisation of less constructive coping strategies, and the deployment of more maladaptive coping strategies were associated with elevated symptoms of anxiety and depression. Similarly, [5] ([5]) observed a positive correlation between resilience and positive affect, as well as subjective well-being. Consequently, individuals who demonstrate resilience in the face of adversity tend to report higher levels of self-esteem and happiness, in contrast to those who lack resilience. Both positive mental health and subjective well-being can be regarded as both essential factors of resilience and consequences of the utilisation of resilient strategies in adverse situations.

In terms of external factors, the most extensively researched factor in resilience is social support. The beneficial effects of formal support provided by institutions and informal support from relatives, partners, and friends have been documented in both quantitative and qualitative studies ([17]; [31]). The perception of social support has been identified as a pivotal resource for families with children diagnosed with Autism Spectrum Disorder (ASD). This perception has been found to be strongly associated with lower maternal stress, reduced instances of depression and anxiety, an enhanced quality of life, and elevated levels of family functioning ([28]; [12]). [25] ([25]) discovered that the levels of social support exerted a moderating influence on the relationship between behavioural issues and the well-being of mothers with children who have intellectual and developmental disabilities. This finding underscores the significance of social support as an important factor.

While previous studies have explored various promoting factors associated with resilience in parents of children with ASD, there exists a significant gap in our understanding of how these factors interact in the post-pandemic context. Specifically, there is a scarcity of research that simultaneously examines the interaction between subjective well-being, positive mental health, social support, humour, cognitive reappraisal, and self-esteem in relation to resilience in this population. Additionally, the integration of emotional intelligence into these frameworks offers a promising avenue for exploring how adaptive emotional skills can further enhance resilience, particularly in high-stress environments. Moreover, little is known about the relative strength of these relationships and how they may have been impacted by the experiences of the pandemic. This study seeks to address this gap, providing a more nuanced understanding of the factors contributing to long-term resilience in parents of children with ASD in the post-pandemic landscape.

The principal aim of this study is to ascertain whether promoting variables, including subjective well-being, positive mental health, social support, humour, cognitive reappraisal, and self-esteem are associated with resilience in parents of children with ASD. In light of the aforementioned considerations, two specific objectives have been formulated as follows: To analyse the capacity of cognitive factors (subjective well-being, positive mental health, social support, humour, cognitive reappraisal, and self-esteem) to modulate resilience in parents of children with ASD, and to analyse the strength of the direct relationship between these factors and resilience.

The initial hypotheses are twofold. First, participants who score high on promoting factors (subjective well-being, positive mental health, social support, humour, cognitive reappraisal, and self-esteem) will also score high on resilience. Second, the highest relationship with resilience will be found in factors such as social support and self-esteem, with humour playing a somewhat lesser role.

## 2. Materials and Methods

### 2.1. Participants

The sample consisted of 799 participants. Inclusion criteria were as follows: (1) being 18 years or older, as the study targeted adult caregivers capable of providing informed consent; (2) having a child with confirmed ASD diagnosis, as this population was central to the study’s focus on the relationship between caregiving and parental resilience, coping, and well-being; (3) completing all questionnaires, to ensure full data collection for analysis; (4) accepting and signing informed consent. Exclusion criteria were: (1) parents with severe psychiatric conditions that might interfere with their ability to reliably participate in the study; (2) children with other pathologies or other comorbid conditions alongside ASD. Of the sample obtained, 53% were female and 47% were male, aged between 26 and 70 years (M = 36.79; SD = 11.81). Regarding cases in which a family member had COVID-19 symptoms, 7% were found to have COVID-19. In addition, 24% lived in dwellings between 60 and 99 square metres. Subsequently, the total sample was divided into two groups, high resilience (M = 81; ST = 12.15) and low resilience (M = 38; SD = 4.97), using the [69] ([69]) Resilience Scale (RS-14) described in the assessment tools and using 64 as the cut-off point, (above the normal group). Table 1 shows the socio-demographic characteristics of the entire sample. There were no significant differences in most of the demographic variables. In terms of effect size, the ETA squared ranged from 0.29 to 0.84.

### 2.2. Evaluation Measures

An ad hoc questionnaire with the variables gender, age, educational level, employment status, presence of COVID-19 infection, and type of housing was used.

The Psychological Well-being Scale (PBS) developed by [68] ([68]), specifically the subjective well-being subscale, was used. It measures satisfaction with life and positive and negative affect. The higher the score, the higher the subjective perception of well-being. The full scale consists of 65 items divided into four subscales. The subscale used in this study consists of 10 Likert-type items with scores ranging from 1 (never or almost never) to 5 (always), with scores closer to 50 indicating subjective well-being. In this sample, a Cronbach’s alpha of 0.75 was obtained.

The Positive Mental Health Scale (PMH scale) by [43] ([43]), applied to the general Spanish population by [79] ([79]), was used. It allows for the evaluation of emotional, psychological, and social well-being through a single measurement. It consists of 9 Likert-type items, where the participant has to select the degree of agreement with the different statements from 1, ‘not true’, to 4, ‘true’. The range of scores is from 9 to 36, with scores closer to 36 being those that would identify participants as having more positive mental health. In terms of reliability, the internal consistency of the original scale, as measured by Cronbach’s alpha, is 0.93, and in the present sample the total alpha was 0.87.

The Coping Humour Scale (CHS-5), originally proposed by [45] ([45]) and translated into Spanish by [9] ([9]), consists of a unidimensional Likert scale ranging from 1 ‘strongly disagree’ to 4 ‘strongly agree’. Scores range from 4 to 20, with higher scores indicating greater coping through humour. The Cronbach’s alpha in this sample is 0.75.

The Reappraisal Index ([22]) in its nine-item version was used. This scale has three dimensions: learning/growth (items 1 and 2), reappraisal of effort (items 3 and 4), and reframing, defined as looking for the positive (items 6, 7, 8 and 9). The Likert scale ranges from ‘1 = never/not at all’ to ‘5 = always/very often’. The direct sum of the scores would make it possible to distinguish between participants who are able to make a positive cognitive reappraisal of various stressful events in their lives and those who do not make such a reappraisal. The internal consistency of the sample is 0.83.

The Rosenberg Self-Esteem Scale, proposed by [65] ([65]) and translated by [14] ([14]), measures personal self-esteem, understood as feelings of personal worth and self-respect. It consists of ten items, five of which are direct, positive, and the last five are inverse, negative. It is a four-point Likert scale ranging from ‘strongly agree’ to ‘strongly disagree’. The result of the sum of the scores obtained classifies the participant into three possible levels: high self-esteem (30 to 40 points), medium self-esteem (26 to 29 points), and low self-esteem (below 25 points). This measure has a Cronbach’s alpha of 0.87 ([78]).

We used the Lubben Social Network Scale (LSNS) ([41]) in its reduced version of six items ([42]) translated into Spanish ([23]). It assesses the social support perceived by participants from their friends and family members through two dimensions: the first, related to family members, made up of the first three items, and the second, related to people with whom they have a relationship but who are not family members, made up of the remaining items. They are in six-point Likert format, where ‘0 = 0 relatives/friends’ and ‘6 = more than nine relatives/friends’. Scores range from 0 to 30, with higher scores indicating greater perceived social support. Cronbach’s alpha is 0.82.

The Resilience Scale (RS- 14) designed by [80] ([80]), whose adaptation to the Spanish population was carried out by [69] ([69]), was used. The original scale measures the level of resilience through two dimensions: personal competence, consisting of 11 items, and acceptance of oneself and life, with three items. The items are presented in a seven-point Likert format, where ‘1 = strongly disagree’ and ‘7 = strongly agree’. The final score obtained allows the participant to be categorised into five levels of resilience: very high resilience (98–82), high resilience (81–64), normal (63–49), low (48–31), and very low resilience (30–14). The Cronbach’s alpha for the current sample is 0.94.

### 2.3. Design and Procedure

This is a non-experimental, descriptive and correlational study with a cross-sectional design. First, the documentation was completed to obtain the approval of the Ethics Committee of the University of Jaén (Spain) and to guarantee the confidentiality of the data provided by the participants.

Secondly, once the favourable report was received, data collection began. All data collection was carried out online using Google Forms by the University of Jaén. The convenience sampling strategy was used, which consists of forming the sample from the cases available through social networks.

### 2.4. Data Analysis

First, descriptive analyses of the scores of all the instruments are analysed, looking at the mean and standard deviation, in order to determine the profile of the total sample. In addition, the assumptions of normality and homogeneity of variance were checked. Then, Student’s *t*-test was used to test for possible statistically significant differences between the scores obtained by the High Resilience Group (HR) and the Low Resilience Group (LR) on the different instruments. We then tested whether there was a correlation between the different variables using Pearson’s correlation coefficient. The significance level is used to determine whether the relationship between the variables is statistically significant. Then, the correlation coefficient (r) is used to determine the direction of the relationship through the statistics, as well as the strength of the relationship, using the following rule: 0.10 to 0.29 low correlation, 0.30 to 0.49 medium correlation, and 0.50 to 1 high correlation. ([46]). The practical implications of the correlations found are also considered. Low correlations (0.10 to 0.29) suggest that while there may be a statistically significant relationship between variables, the effect may be small, and thus not always impactful in clinical settings. Medium correlations (0.30 to 0.49) may suggest that variables such as emotional regulation or mental health have a more substantial but still context-dependent impact on resilience. In clinical practice, these moderate correlations indicate that targeted interventions focusing on these variables could be beneficial but should be considered alongside other factors. This understanding of both statistical and practical significance helps guide the development of interventions aimed at improving well-being in this population. High correlations (0.50 to 1) indicate a strong relationship between variables. This level of correlation would provide robust evidence to guide the development of focused interventions. For instance, enhancing mental health strategies could lead to marked improvements in resilience among parents, potentially offering long-term benefits for both the caregivers and their children. High correlations are particularly relevant when seeking to identify key intervention targets in psychological care.

The reliability of the instruments included was also measured using Cronbach’s alpha coefficient. Finally, a stepwise multivariate regression analysis is performed for a predictive model, first calculating goodness-of-fit indices for sociodemographic and some variables such as positive mental health, emotional regulation, CHS-5 score, and social support, these being the independent variables and resilience being the dependent variable. To address potential multicollinearity among the independent variables, Variance Inflation Factor (VIF) values were calculated. A VIF value below 10 was considered acceptable, indicating that multicollinearity was not a major issue in the model. Statistical power and effect size were also calculated. The required level of statistical significance in the tests was a minimum of *p* < .05. Regarding the interpretation of goodness-of-fit indices, R-squared (R^2^) was used to assess the proportion of variance explained by the model, with higher values indicating better fit. Adjusted R^2^ was also reported to account for the number of predictors in the model. Additionally, F-statistics were used to assess the overall significance of the model. These indices provide insight into the model’s predictive power and its relevance for understanding the predictors of resilience in parents of children with ASD. Statistical analysis was performed using the SPSS statistical package version 22.0 ([34]), and statistical power and effect size were calculated using G*Power 3.1.9.7. ([16]).

## 3. Results

The descriptive analysis data (Table 2) show that psychological well-being is moderate. Comparing the scores between the High Resilience (HR) and Low Resilience (LR) groups, we see that there are significant differences, with the average score of the HR being higher than that of the LR. In terms of positive mental health, statistically significant differences were observed between the two groups, with the HR scoring higher than the LR. With regard to humour, the mean scores obtained indicate a high use of humour as a coping mechanism, but there are no significant differences between the two groups. Parents of children with ASD report high levels of cognitive reinterpretation, with scores ranging from 18 to 45. When the means of the groups are examined, statistically significant differences are observed, with the mean scores of HR being higher than those of LR. The score obtained on the self-esteem scale indicates a medium level of self-esteem, with 59.3% of the participants reporting high self-esteem, compared to 29.6% with low scores, with no significant differences between the two groups. Finally, with regard to social support, the range of scores obtained was between 2 and 30, with the mean of the population slightly above the proposed cut-off point, and there were no differences between the two groups of high and low resilience.

Secondly, correlations were established between the different variables and resilience to determine their relationship (Table 3). Statistically significant correlations are observed between resilience and the variables psychological well-being, positive mental health, cognitive reinterpretation, and self-esteem. In addition, the variable cognitive reinterpretation correlates significantly with psychological well-being, positive mental health, and sense of humour. Significant correlations are also observed between psychological well-being, positive mental health, and sense of humour.

Finally, high reliability is observed as measured by Cronbach’s alpha for resilience 0.954 as well as the highest reliability for Positive Mental Health and Cognitive Reassessment.

Finally, a multiple regression analysis was performed to examine which psychological and socio-demographic variables were most predictive of resilience in a sample where family members with a child with ASD (*N* = 799) experienced this due to the COVID-19 pandemic confinement policies.

Preliminary analyses for goodness-of-fit assessment confirmed that the assumptions of non-multicollinearity (<5, PIV = 1.00 and 1.77; [37]) were met and the tolerance values (1–0.1) were between 1 and 0.98. Furthermore, there was no autocorrelation in any of the psychological and sociodemographic variables, so the assumption of error independence (Durbin–Watson = 1–3) was correct and the results can be generalised to the general population, with the coefficient close to two (DW = 1.95) ([83]). Therefore, a stepwise multiple regression (explanatory variables enter the model according to their degree of correlation with the dependent variable, in this case resilience) was performed to detect the level of significance of each socio-demographic and psychological variable in order to detect the most appropriate and best-fitting prediction model for this sample (Table 4).

Some sociodemographic and psychological variables explain resilience to a greater degree, with the proposed model (set of independent variables) being significant and explaining 73.4% of resilience in this sample (R2c = 0.734; *F*(1,798) = 537.01; *p* < .01). The last proposed model, composed of sociodemographic and psychological variables (model 3) would indicate the variables that predict resilience to a greater extent.

The results referring to socio-demographic variables show that specifically the 48 to 58-year-old group (β = 3.03; CI (95%) = 2.12–3.74; *p* < .01) and the self-employed group (β = 1.17; CI (95%) = 1.07–3.42; *p* < .01) are the socio-demographic variables that would best explain a higher level of resilience. Regarding the other variables, the data show positive mental health (β = 9.13; CI (95%) = 8.01–9.22; *p* < .01), emotion regulation (β = 8.36; CI (95%) = 8.14–8.79; *p* < .01), sense of humour, measured by the CHS-5 score (β = 6.11; CI (95%) = 5.26–7.63; *p* < .01), and social support (β = 9.28; CI (95%) = 8.61–10.01; *p* < .01) are variables that predicts resilience.

Thus, incorporating variables into the model generates greater power to explain resilience from our independent variables, having a high level of statistical power (1 − β = 1) and effect size (f2 = 20.1) ([49]).

## 4. Discussion

The aim of the present study was to determine whether the psychological variables assessed (subjective well-being, positive mental health, social support, sense of humour, cognitive reappraisal, and self-esteem) were related to resilience in parents of children with ASD. As mentioned above, parents of children with ASD are a particularly vulnerable population, given the risk that COVID-19 poses to their children, in addition to their increased caregiver burden due to limited support networks ([1]; [7]; [74]).

Based on the results obtained, the established working hypotheses are confirmed. Firstly, those participants who scored high on the promoting factors examined also scored high on resilience. Regarding the second hypothesis, it was confirmed that the promoting factor to which resilience was least related was humour; however, the strongest relationship with resilience was not with social support and self-esteem, but with psychological well-being, cognitive reappraisal and positive mental health.

These findings are consistent with previously reviewed studies, such as that of [5] ([5]), which highlights the relationship between psychological well-being and positive mental health, and how these variables promote resilience in those who develop a caring role. Another study that supports this finding is that of [8] ([8]), which highlights the important relationship between subjective well-being and resilience, particularly for suicide prevention.

With regard to cognitive reappraisal, the results obtained are in line with studies such as that of [10] ([10]), in which variables such as cognitive reappraisal were highlighted as a characteristic of resilient behaviour in the sample following the implementation of a programme to promote resilience among informal carers. This finding is also supported by studies such as that of [54] ([54]), in which cognitive reappraisal was directly related to resilience. Cognitive reappraisal is a key process within emotional intelligence (EI), as it allows individuals to regulate their emotions by reframing negative situations in a more constructive way ([13]; [50]). In line with this, it is crucial to further explore the relationship between resilience and self-regulation, as highlighted in the recent literature ([47]). Self-regulation, encompassing cognitive and emotional processes such as cognitive reappraisal, is fundamental to resilience because it allows individuals to adapt flexibly to changing circumstances and manage stress effectively ([76]). This aligns with our findings, where cognitive reappraisal emerged as a significant predictor of resilience in parents of children with ASD. Emotional intelligence frameworks, which include self-regulation and cognitive flexibility, appear to be crucial for fostering resilience in high-stress caregiving contexts ([44]).

The present study also analysed the relationship between self-esteem and the variables studied (cognitive reappraisal and resilience), and the results indicated that self-esteem only partially mediated the relationship between cognitive reappraisal and resilience, which is consistent with the results of the present study and the weak relationship between resilience and self-esteem.

One of the promoting variables measured and analysed in this research was positive mental health, which was significant in predicting resilience, concordant with other studies that have highlighted the emotional component in the positive mental health scale ([33]; [73]; [77]). Positive mental health is conceptually linked to emotional intelligence, as it reinforces self-efficacy and emotional self-regulation, both of which are critical for fostering resilience in challenging contexts ([47]). Emotional regulation has also been significant in predicting resilience. In line with other studies, for example, in young samples ([54]) or a systematic review by [58] ([58]), confirmed that emotion regulation can facilitate emotion and problem-focused coping, promoting psychological resilience. The regulation of emotions, a key facet of emotional intelligence, contributes to resilient behaviours by enhancing emotional balance and adaptive coping skills, which are crucial for navigating caregiving challenges ([56]). By incorporating self-regulation strategies, such as reframing stressful caregiving experiences positively, caregivers can enhance their emotional balance and problem-solving capacity, both of which are integral to resilient outcomes ([67]). This underscores the potential value of interventions aimed at strengthening self-regulation and emotional intelligence to bolster resilience in this population.

Sense of humour was also an important variable in predicting resilience, according to the works of [75] ([75]), who found that the variables of sense of humour and life satisfaction significantly predicted psychological resilience in teacher candidates. Humour, while less directly related to resilience, may still play a role in emotional intelligence by supporting emotional regulation, reducing stress, and promoting adaptive coping strategies ([84]). The last promoting variable examined was social support, which also significantly predicted resilience. [71] ([71]) also found that there was a positive and significant relationship between social support and resilience and the quality of life of parents of disabled children. Other studies also align with these findings ([19]; [27]; [85]).

Continuing with the variables that predict resilience, we found that the main socio-demographic variables in our target population were age (specifically, the age group of 48 to 58 years) and being self-employed, which is in line with previous research. Specifically, in the case of age, the older the parent, the higher the level of resilience ([39]; [61]). Being self-employed or employed, having a higher level of education and living in a larger dwelling are facilitating variables ([6]; [24]; [36]).

Psychological well-being and positive mental health foster resilience through several psychological mechanisms. One potential mechanism is cognitive flexibility, which allows individuals to adapt their thinking patterns in response to adversity, facilitating more effective problem-solving and emotional regulation ([18]). Additionally, individuals with higher levels of psychological well-being tend to engage in more adaptive coping strategies, such as cognitive reappraisal, which involves reframing negative situations in a more positive light ([63]). These mechanisms are integral components of emotional intelligence, which emphasises adaptability and emotional balance in challenging circumstances. This reframing process reduces stress and helps maintain emotional balance, thereby enhancing resilience. Positive mental health also reinforces a sense of self-efficacy, enabling individuals to believe in their ability to overcome challenges, which further strengthens resilience ([35]). These mechanisms suggest that interventions aimed at improving psychological well-being and mental health could have a significant impact on increasing resilience in parents of children with ASD ([72]).

Although the relationship between humour and resilience was weaker than expected, this does not necessarily mean humour lacks relevance in the context of coping. Humour may still serve as a personal resource for managing stress ([21]), but it appears to have a less direct influence on resilience in this specific population. Humour may instead contribute indirectly by enhancing emotional intelligence components such as emotional regulation or interpersonal skills, which could support resilient behaviours over time. It is possible that, given the chronic and high-stress nature of caregiving for children with ASD, other mechanisms, such as cognitive reappraisal or psychological well-being, may play a more prominent role ([15]).

It is also important to note that resilience manifests differently across populations. While humour may be more impactful in settings with moderate or temporary stress, it may not be as effective in highly demanding and prolonged caregiving contexts. This opens the door for future research to explore whether humour has indirect effects on resilience by influencing other dimensions, such as emotional regulation or life satisfaction, which might, in turn, support resilient behaviours ([29]).

The data obtained in this study highlight the influence of psychological well-being and positive mental health on the level of resilience in the population studied. These findings underscore the relevance of emotional intelligence and self-regulation processes, which enable caregivers to maintain emotional balance and adapt to stressors, thereby promoting resilience. This evidence supports the importance of the correct psychological functioning of individuals to facilitate coping with adverse situations that generate a significant impact. Resilient coping reduces the possible psychological consequences and facilitates coping with these situations. When designing intervention programmes with parents of children with ASD, we need to consider the general psychological health of the individual as a crucial factor, particularly as a preventive mechanism against the possible psychological sequelae that may result from their children’s disability.

In terms of practical implications, for example, incorporating resilience-building programmes that include cognitive reappraisal and emotional regulation techniques could be beneficial for this population. Additionally, focusing on providing social support systems tailored to the needs of caregivers could strengthen their capacity to cope with stress. Integrating emotional intelligence frameworks into these programmes could further enhance their effectiveness by promoting skills such as emotion regulation, interpersonal understanding, and adaptability. This could be achieved through peer support groups, structured caregiver counselling, or community-based interventions designed to enhance access to resources and support. These approaches would help mitigate the caregiver burden and promote long-term psychological resilience. The multisystemic perspective of resilience further suggests that effective self-regulation not only supports individual resilience but also contributes to the resilience of interconnected systems, such as family and community networks. Further studies might investigate the dynamic interplay between self-regulation processes and resilience at these broader levels, particularly in populations facing prolonged stressors like those of caregivers to children with ASD.

Future research should also explore the development of resilience over time in this population. A longitudinal approach would be valuable in understanding how resilience evolves as parents face ongoing challenges in caring for children with ASD. Tracking changes in resilience and identifying critical periods where interventions may be most effective could improve long-term outcomes. Furthermore, mixed-methods research that integrates qualitative data could provide deeper insights into the personal experiences and coping mechanisms of parents, complementing quantitative findings and enriching our understanding of resilience in this context.

## 5. Limitations

There are several limitations to this study. On the one hand, emotional disorders such as depression and anxiety, which may mediate and condition the results obtained in terms of positive mental health, were not taken into account. In this vein similar studies have introduced optimism which, in addition to being positively related to resilience in other studies ([57]; [64]), would have been interesting to include in this work. Other socio-demographic and psychological variables such as education level, socioeconomic status, family support, and prior mental health conditions that could affect the Spanish population during incarceration would also be useful to consider.

Additionally, the sampling method used, based on convenience sampling through social networks, could have introduced biases into the sample. Participants recruited through social networks may not be fully representative of the general population of parents of children with ASD, potentially affecting the generalizability of the results. This method may have led to the overrepresentation of certain groups, such as individuals who are more active on social media or more engaged with online communities. These biases could have influenced the findings, particularly regarding the psychological variables under study, such as resilience and mental health.

On the other hand, while the cross-sectional design used provides important insights, it limits the ability to draw conclusions about causality. A longitudinal design could provide a deeper understanding of how the variables interact over time, allowing us to observe potential changes in resilience and mental health outcomes in parents of children with ASD as they face ongoing challenges. Tracking these variables longitudinally would help identify whether certain promoting factors, such as optimism or social support, influence resilience in a sustained way or only during specific periods of stress. Additionally, future research could benefit from employing mixed-method approaches and integrating both quantitative and qualitative data. This would offer a more comprehensive view of resilience by capturing not only measurable outcomes but also the personal experiences and coping strategies of parents, which may provide deeper insights into how they adapt to the demands of caring for a child with ASD. Such methodologies could complement traditional statistical analyses and enrich the understanding of resilience in this population. In addition, the questionnaires were self-administered, which leads to more subjectivity.

Regarding the statistical analyses, latent variable modelling could provide a more robust estimation of the relationships by accounting for measurement error and improving the reliability of constructs. However, we did not employ this approach in the current study due to the complexity of such analyses with the number of variables and the limitations in terms of the theoretical framework of the study. We also opted for a more direct method of analysis in line with the study’s goals, aiming to provide more practical and interpretable results within the scope of the research. However, future research with the same sample size could certainly benefit from latent variable modelling to gain deeper insights into the relationships between the variables.

## Figures and Tables

**Table 1 jintelligence-13-00006-t001:** Description of the socio-demographic data of the total sample and sub-samples.

	N (%)	HRn (%)	LRn (%)	Z	η^2^
Gender					
Woman	421 (52.7)	215 (51.1)	206 (48.9)	−0.129 ^ns^	0.60
Man	378 (47.3)	169 (44.7)	209 (55.3)		
Age					
26–36	299 (37.4)	130 (43.5)	169 (56.5)		
37–47	247 (30.9)	118 (47.8)	129 (52.2)	−0.150 ^ns^	0.29
48–58	124 (15.6)	58 (46.8)	66 (53.2)		
59–70	129 (16.1)	61 (47.3)	68 (52.7)		
Academic Level					
None	97 (12.1)	18 (18.6)	79 (81.4)		
Secondary School	243 (30.4)	119 (49)	124 (51)	3.02 **	0.84
University/Voc. training	357 (44.7)	176 (49.3)	181 (50.7)		
Postgraduate	102 (12.8)	58 (56.9)	44 (43.1)		
Employment situation					
Active	357 (44.7)	188 (52.7)	169 (47.3)		
Freelance	194 (24.3)	99 (51)	95 (49)	5.22 ^ns^	0.59
Retired	58 (7.3)	18 (31)	40 (69)		
Unemployed/ERTE/ERE	190 (23.8)	102 (53.7)	88 (46.3)		
Presence of COVID-19 infection					
Yes	52 (6.5)	17 (32.7)	35 (67.3)	1.09 **	0.68
No	747 (93.5)	359 (48.1)	388 (51.9)		
Type of housing					
Flat ≤ 59 m^2^	178 (22.3)	86 (48.3)	92 (51.7)		
Flat 60–99 m^2^	188 (23.5)	91 (48.4)	97 (51.6)		
Flat ≥ 100 m^2^	177 (22.2)	86 (48.6)	91 (51.4)	1.49 ^ns^	0.37
One-storey house 100 m^2^	154 (19.3)	75 (48.7)	79 (51.3)		
Two-storey house 100 m^2^	102 (12.8)	50 (49)	52 (51)		

*Note.* HR = High resilience group, LR = Low resilience group; Z = Man Whitney’s U; ns = no significant; ** significant correlation at the 0.01 level (bilateral); η^2^ = effect size.

**Table 2 jintelligence-13-00006-t002:** Descriptive results by variable.

Variable	HR		LR		*t*	*p*
	*M*	*SD*	*M*	*SD*		
PW-B	34.35	2.74	26	3.33	3.01	0.001
PMH	30.9	3.02	24	2.66	1.76	0.001
CHS-5	13.52	2.08	11.04	1.71	0.23	0.120
CR	33.05	3.05	20.16	2.21	2.01	0.002
S	27.23	3.29	24.8	2.04	0.81	0.384
SS	14.95	3.06	12.16	3.75	1.06	0.242

*Note.* HR = High resilience group; LR = Low resilience group; P-WB = Psychological Well-Being; PMH = Positive Mental Health; CHS-5 = Sense of humour; CR = Cognitive Re-evaluation; S = Self-Esteem; SS = Social Support; M = Mean; SD = Standard Deviation; *t* = Student’s *t*; *p* = significance.

**Table 3 jintelligence-13-00006-t003:** Correlations with resilience and alpha reliability of all measures.

	P W-B	PMH	CHS-5	CR	S	SS	R	α
P W-B	1	0.76 **	0.52 **	0.68 **	−0.09	0.25	0.67 **	0.753
PMH	0.76 **	1	0.27	0.41 *	0.33	0.28	0.61 **	0.871
CHS-5	0.52 **	0.27	1	0.61 **	−0.15	−0.28	0.25	0.752
CR	0.68 *	0.41 *	0.61 **	1	0.20	−0.17	0.66 **	0.831
S	−0.09	0.33	−0.10	0.30	1	0.06	0.37 *	0.827
SS	0.25	0.21	−0.028	−0.17	0.01	1	0.28	0.826
R	0.67 **	0.61 **	0.25	0.66 **	0.37 *	0.28	1	0.954

*Note.* P W-B = Psychological Well-Being; PMH = Positive Mental Health; CHS-5 = Sense of humour; CR = Cognitive Re-evaluation; S = Self-Esteem; SS = Social Support; R = Resilience; α = Cronbach’s alpha; ** correlation significant at the 0.01 level; * correlation significant at the 0.05 level.

**Table 4 jintelligence-13-00006-t004:** Predictive models of resilience with socio-demographic and promoting variables in the sample (n = 799).

Models and Variables	R^2adj^	SE	*F*	*t*	β	CI (95%) (β)		*φ* ^2^
						L.L.	U.L.	
Step 1	0.230	1.01	12.11 ^ns^	0.12 ^ns^				0.30
Age (48–58)					0.51	−0.10	1.01	
Self-employed worker					0.32	−0.06	1.15	
Social Support					0.48	−0.11	1.10	
Step 2	0.607	1.37	120.1 **	3.86 *				10.2
Age (48–58)					2.35	−0.10	3.55	
Self-employed worker					2.43	−1.11	2.89	
Positive mental health					2.24	0.21	2.41	
Emotion regulation (cognitive reappraisal)					2.62	1.23	2.84	
CHS-5					1.01	1.24	1.71	
Social Support					3.08	1.13	4.61	
Step 3	0.734	2.25	537.01 **	53.23 **				20.1
Age (48–58)					3.03	2.12	3.74	
Self-employed worker					1.17	1.07	3.42	
Positive mental health					9.13	8.01	9.22	
Emotion regulation (cognitive reappraisal)					8.36	8.14	8.79	
CHS-5					6.11	5.26	7.63	
Social Support					9.28	8.61	10.01	

R^2adj^ = Adjusted R-squared; SE = standard error; F = test statistic (ANOVA); * *p* < 0.05 ** *p* < 0.01; ns = non-significant; *t* = predictive variable test statistic; β = result of regression or beta equation; CI 95% = confidence intervals; LL = lower limit; UL = upper limit; *φ*^2^ = effect size; CHS-5 = sense of umor.

## Data Availability

The data presented in this study are available upon request from the corresponding author.

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
