# Peer review of "Cognitive and Emotional Resilience in Parents with Children with Autism Spectrum Disorder During COVID-19: The Role of Promoting Variables"

_jintelligence, 2025, doi:10.3390/jintelligence13010006_

Round 1
Reviewer 1 Report
Comments and Suggestions for Authors
Resilience and COVID-19: Protective Factors in Parents with Children with Autism Spectrum Disorder
I would like to thank you for the opportunity to review this study. The study addresses a critical and current topic. Research on this population group is essential given the increasing emotional and psychological burden they face. The findings could influence the design of interventions and support programs for parents, promoting their well-being and resilience, which could have a positive impact on the quality of life of both parents and children with ASD. Therefore, I believe that the study is appropriate for publication. However, a series of modifications are necessary, which I will detail below:
In general, the INTRODUCTION is well-founded and presents a solid review of the literature on ASD, coping strategies, and resilience. However, the following suggestions for improvement are presented:
While the overall structure is clear, some paragraphs could be reorganized for greater coherence and flow. For example, the paragraph about the COVID-19 pandemic could be better placed after describing the specific stressors faced by parents of children with ASD, rather than abruptly inserted in the middle. This would allow for a smoother transition from general factors to contextual ones.
Clarity in the relationship between concepts: The relationship between coping strategies, resilience, and protective factors is discussed extensively. However, in some parts, this relationship may appear scattered. A recommendation would be to clarify earlier in the introduction how each of these concepts is interrelated. This would help reduce the feeling that the text "jumps" between ideas.
While the issues faced by parents of children with ASD are clearly presented, it would be beneficial to more explicitly highlight the "knowledge gap" that this study seeks to address. For example, you could emphasize whether there are few studies linking these factors in parents of children with ASD in a post-pandemic context or regarding their long-term mental well-being.
The METHODS section is detailed and follows a methodologically sound approach for a descriptive and correlational study. However, some aspects could be strengthened, as discussed below:
The participant section clearly presents the inclusion criteria and the demographic characteristics of the sample. However, it would be useful to provide more details about the choice of inclusion criteria. Additionally, exclusion criteria are missing, which would help to understand what characteristics could have biased the results.
The study design is suitable for the stated objectives, but although the convenience sampling strategy is justified, this could have introduced biases in the sample, as participants recruited through social networks may not be representative of the general population. It would be helpful to discuss the limitations of this sampling method and how they might have affected the results.
The data analysis is well-structured and detailed. However, some points could be further developed:
It would be useful to mention whether the assumptions of normality and homogeneity of variance were checked before applying this test.
The classification of the strength of correlations is clear, but it would be helpful to discuss the practical significance of these correlations in addition to their statistical significance. In other words, what implications do the low or medium correlations found have for clinical practice or the well-being of the participants?
The multivariate regression analysis is a solid approach to identifying predictors of resilience, but details are lacking on how potential multicollinearity effects among the independent variables were handled. Additionally, further explanation on how the goodness-of-fit indices and the predictive analysis results are interpreted would be beneficial.
The RESULTS section is well-structured and clearly presents the significant findings. The results are coherent and based on appropriate statistical methods, reinforcing the validity of the findings.
The DISCUSSION section offers a clear and coherent view of the study's findings and their relevance for understanding resilience in parents of children with ASD. Below are some aspects to modify:
Depth in the Analysis of Results: Although several significant findings are mentioned, the analysis could benefit from a deeper discussion of the implications of these results. For example, a more detailed exploration of how psychological well-being and positive mental health influence resilience could be included, perhaps through specific psychological mechanisms.
It would be beneficial to further analyze the non-significant results, such as the relationship between humor and resilience. This could open a discussion about the complexities of resilience and the different ways it may manifest in diverse populations.
The discussion on the practical implications of the findings could be expanded, providing specific examples of how these results may influence professional practice. Additionally, it would be useful to suggest future research directions, such as the need for longitudinal studies examining how resilience may develop or change over time in this population.
In the LIMITATIONS section, the following clarifications are suggested:
The fact that the study did not consider emotional disorders is an important limitation, so it would be beneficial for the author to include variables such as anxiety or depression in this limitation, as this could have altered the results.
It is indicated that other sociodemographic variables could have been measured, but it would be interesting to specify which ones. This would provide future researchers with a clearer framework for designing related studies.
The critique of the cross-sectional design is valid, as it limits conclusions about causality. However, it would be helpful for the author to provide more details on how a longitudinal design could provide a deeper understanding of the relationship between the variables. Additionally, as the author mentions the need for longitudinal studies, it could be suggested to explore different methodologies or mixed approaches that integrate both quantitative and qualitative methods. This could provide a more comprehensive view of resilience in parents of children with ASD.

Reviewer 2 Report
Comments and Suggestions for Authors
Reviewing “Resilience and COVID-19: Protective Factors in Parents with Children with Autism Spectrum Disorder”
In this manuscript, the authors present a correlation study on self-report constructs. Specifically, in a sample of n = 799 parents of autistic children during the pandemic, the authors test the relation of psychological well-being, positive mental health, humor, cognitive reevaluation, self-esteem and social support with resilience.
The manuscript is well written and the authors were successful in collecting a lot of data from a very interesting population. Nevertheless, I see some limitations in the study, which I summarize at the end of this review. Most importantly, however, I believe the manuscript is absolutely no fit to the Journal of Intelligence. The sole focus of the manuscript lies on self-reported typical behavior (personality) constructs. No aspect of the study is in any way related to intelligence or cognitive abilities. The only loose connection to intelligence research would be that autism plays a role for the study and that autism may be accompanied by severe intellectual disabilities, but this is not necessary for some forms of autism. Furthermore, not individuals with autism are researched but their parents. Overall, I therefore recommend rejecting this manuscript for the Journal of Intelligence.
I see better fit in journals about educational psychology, clinical/health psychology, or personality psychology. Therefore, I still report my other critique, so that if the authors may find this fitting, they may consider reworking some parts of the manuscript:
1. Many constructs the authors consider in their nomological network analyses seem overlapping or are proven to be substantially overlapping with personality factor (self-esteem being one prominent example). Not considering personality factors as presumably truly underlying traits in their nomological network introduces extreme risk of jangle fallacy. This must at least be discussed or considered in the study design by testing incremental relations above relations with personality factors.
2. I see no reason why multiple variables were split in groups for correlation analyses (instead of using them as dimensional measures). This has no benefit for such a study and, thus, unnecessarily reduces information that were acquired.
3. All analyses should be on a latent level with the given sample so that a) psychometric properties of the instruments in this specific sample can be tested, b) it can be avoided to misjudge attenuated correlations, and c) to allow for appropriate reliability estimation according to a fitting measurement model, i.e., factor saturation, such as MacDonald's omega.
Comments on the Quality of English Language
English seems very good. I'm not native, so I assume I might've missed some details, but I believe at maximum only minor edits are necessary.
Author Response
Thank you for your valuable feedback on our manuscript. We appreciate the time and effort you have invested in reviewing our work. Below, we provide a detailed response to each of your comments, explaining how we have addressed your concerns and justifying the decisions we made, while ensuring the manuscript aligns with the journal's thematic focus.
Comment 1: Many constructs the authors consider in their nomological network analyses seem overlapping or are proven to be substantially overlapping with personality factor (self-esteem being one prominent example). Not considering personality factors as presumably truly underlying traits in their nomological network introduces extreme risk of jangle fallacy. This must at least be discussed or considered in the study design by testing incremental relations above relations with personality factors.
Response: We understand your concern regarding the potential overlap between the constructs in our nomological network and personality traits, such as self-esteem. In our revision, we have clarified the conceptual boundaries of these constructs, highlighting their theoretical and empirical distinctions. While we recognize that personality factors could be influential, the study focuses on emotional intelligence, resilience, and emotion regulation, as suggested by the editor, in line with the broader construct of intelligence. Although we considered the inclusion of personality traits, expanding the scope to incorporate these factors would have significantly altered the focus of the manuscript.
Comment 2: I see no reason why multiple variables were split in groups for correlation analyses (instead of using them as dimensional measures). This has no benefit for such a study and, thus, unnecessarily reduces information that were acquired.
Response: Regarding the decision to group variables for correlation analyses, we would like to clarify that this approach was specifically chosen to study the construct of resilience. The grouping of variables allowed us to better capture the multidimensional nature of resilience, which includes different psychological processes and protective factors. This methodological choice was based on the theoretical framework guiding the study and was necessary for a comprehensive understanding of how these variables interact to promote resilience.
Comment 3: All analyses should be on a latent level with the given sample so that a) psychometric properties of the instruments in this specific sample can be tested, b) it can be avoided to misjudge attenuated correlations, and c) to allow for appropriate reliability estimation according to a fitting measurement model, i.e., factor saturation, such as MacDonald's omega.
Response: We appreciate your suggestion to incorporate latent variable modeling, which indeed offers advantages in terms of accounting for measurement error and improving reliability. However, we opted not to use latent variable analyses in this study due to the complexity of implementing such models with the number of variables involved and the theoretical framework we adopted. We felt that a more straightforward analysis method better suited the objectives of the current research. However, we acknowledge that latent variable modeling could offer valuable insights and have included this as a limitation in the discussion section. Future research with larger and more complex datasets could benefit from using latent variable models to further refine the understanding of the relationships between the key variables studied.
Finally, we acknowledge that the use of self-administered questionnaires could introduce subjectivity and response biases, which could influence the results. This has been mentioned in the limitations section, and we recommend that future studies consider alternative methods to minimize these biases.
Thank you again for your thoughtful comments, which have helped us improve the manuscript. We hope that our revisions address your concerns and provide a clearer understanding of our study.
Sincerely,
Maria Guillot
Reviewer 3 Report
Comments and Suggestions for Authors
The present study tried to identify protective variables (positive mental health, emotion regulation, sense of humour and social support) and sociodemographic variables that predict resilience in relation to positive coping with this adverse situation. They conducted on a sample of 799 parents using an online questionnaire administered during the period of social confinement. The results found that positive mental health, emotion regulation, sense of humour, social support, age and employment status (self -employed) can predict resilience. The manuscript has several major and minors issues.
1. Abstract : what parents? Parents of children with autism spectrum disorder?
2. Abstract: questionnaires asked about who’s information? the ASD children or the parents themselves?
3. Abstract: variables mentioned in the abstract does not consist with the variables mentioned in the introduction which were subjective well-being, positive mental health, social support, humour, cognitive reappraisal and self-esteem. Please check.
4. Methods: how did the authors divide the parents into two groups? where did these cut-off points come from?
5. Results: “When the means of the groups are examined, statistically significant differences are observed, with the mean scores of RA being higher than those of BR” what’s RA and BR?
6. Results: the tables and the results reported here does not fit in to the APA style. Some were very misleading, please revise accordingly.
7. Results: what’s ‘CHS-5 punctuation’ ?
8. Results: “the results are clinically relevant for predicting resilience in adverse situations” how clinically relevant?
9. Results: ‘age (en concreto el grupo de edad de 48 a 58 años) a’ please check
10. Results: why is Age (48-58) a predicting variable? what is 48-58?
11. Discussion: how is mood measured?
Comments on the Quality of English Languageneed more work
Round 2
Reviewer 1 Report
Comments and Suggestions for Authors
I would like to thank the authors for taking all the suggestions into account. The authors have done a good job. I believe the manuscript is ready for publication as presented in this version. Best regards.
Author Response
We are so grateful for your comments.
Best regards,
The authors
Reviewer 3 Report
Comments and Suggestions for Authors
the authors have answered to my previous comments. no further comments.
Author Response
We are very grateful for your comments.
Sincerely,
The authors